# Formation of Natural Magnesium Silica Hydrate (M-S-H) and Magnesium Alumina Silica Hydrate (M-A-S-H) Cement

**DOI:** 10.3390/ma17050994

**Published:** 2024-02-21

**Authors:** Håkon Austrheim, Depan Hu, Ole Ivar Ulven, Niels H. Andersen

**Affiliations:** 1The Njord Center, Department of Geology, University of Oslo, P.O. Box 1048, Blindern, 0318 Oslo, Norway; oiulven@gmail.com; 2College of Geophysics, Chengdu University of Technology, Chengdu 610059, China; hudepan01@163.com; 3Geoenvironment Monitoring Station in Chengdu, Chengdu 610059, China; 4Observation and Research Station of Chengdu Geological Hazards, Ministry of Natural Resources, Chengdu 610042, China; 5Department of Chemistry, University of Oslo, P.O. Box 1048, Blindern, 0371 Oslo, Norway; n.h.andersen@kjemi.uio.no

**Keywords:** serpentinization, weathering, CO_2_ sequestration, Feragen and Leka ultramafic complexes, brucite, natural (M-(A)-S-H) cement, duricrust

## Abstract

Occurrences of natural magnesium alumina silicate hydrate (M-(A)-S-H) cement are present in Feragen and Leka, in eastern and western Trøndelag Norway, respectively. Both occurrences are in the subarctic climate zone and form in glacial till and moraine material deposited on ultramafic rock during the Weichselian glaciation. Weathering of serpentinized peridotite dissolves brucite and results in an alkaline fluid with a relatively high pH which subsequently reacts with the felsic minerals of the till (quartz, plagioclase, K-feldspar) to form a cement consisting of an amorphous material or a mixture of nanocrystalline Mg-rich phyllosilicates, including illite. The presence of plagioclase in the till results in the enrichment of alumina in the cement, i.e., forms M-A-S-H instead of the M-S-H cement. Dissolution of quartz results in numerous etch pits and negative quartz crystals filled with M-A-S-H cement. Where the quartz dissolution is faster than the cement precipitation, a honeycomb-like texture is formed. Compositionally, the cemented till (tillite) contains more MgO and has a higher loss of ignition than the till, suggesting that the cement is formed by a MgO fluid that previously reacted with the peridotite. The M-(A)-S-H cemented till represents a new type of duricrust, coined magsilcrete. The study of natural Mg cement provides information on peridotites as a Mg source for Mg cement and as a feedstock for CO_2_ sequestration.

## 1. Introduction

Portland cement production contributes c. 8% of the annual anthropogenic CO_2_ release [1], albeit substantial amounts of the CO_2_ is recaptured over time [2]. The technology for producing the magnesium analog by calcination of magnesite (MgCO_3_) has been available for a century but is even more energy consuming and is hampered by high CO_2_ release [3]. The search for a CO_2_-neutral or CO_2_-consuming cement has been focused on the Mg system [3,4] and hydrated magnesium carbonate (HMC) and magnesium silica hydrates (M-(A)-S-H) are two promising candidates. The HMC cement consists of the hydrocarbonates, hydromagnesite, nesquehonite, landsfordite, dypingite and artenite. Hydrocarbonate formation is also reported from many ultramafic complexes [5,6] and, notably, Wilson et al. [7] and Power et al. [8] demonstrate that hydrocarbonate can act as a cement by stabilizing mine tailings. The hydration reaction of magnesium oxide and silica produces magnesium silicate hydrate (M-S-H), which provides binding characteristics like Portland cement binder [9]. The formation of magnesium silicate hydrate has been observed at the interfacial zone of cement-based materials in contact with clays [10] and/or as secondary products from the degradation of cementitious materials by groundwater or seawater [11]. M-S-H cement also form in nature. de Ruiter and Austrheim [12] report tillite cemented by M-S-H and Nisiki et al. [13] describes M-S-H formed on ultramafic rocks by seepage with high activity of Si. Experimentally, the presence of aluminum leads to the formation of M-A-S-H, where Al^3+^ can replace Mg^2+^ in the octahedral sites and Si^4+^ in the tetrahedral positions [14]. The performance and properties of M-S-H cement may be improved by adding Al [15]. Natural M-A-S-H cement has, to our knowledge, not previously been recorded.

Magnesium cement has several advantages over Portland cement in addition to being CO_2_ neutral. It forms at a lower pH than Portland cements and is a promising matrix for storage of nuclear waste [16,17]. According to Bernard [18], binders containing M-S-H also have good mechanical properties, a dense microstructure and potentially good resistance to leaching. Thus, M-S-H is currently of interest as an environmentally friendly alternative to Portland cement due to its potentially lower carbon footprint [3] and for the encapsulation of nuclear waste [19]. The source of MgO is a limiting factor for a large-scale production of Mg cement [4].

De Ruiter and Austrheim [12] discovered and described natural M-S-H from Feragen, where a tillite consisting of clasts of quartz, feldspar and ultramafic material was cemented by a hydrous Mg-silicate cement with an average composition of Mg_8_Si_8_O_20_(OH)_8_·6H_2_O, which is a mixture of nanocrystalline Mg-rich phyllosilicates (e.g., kerolite, stevensite and serpentine). The authors suggested that the cement forms from a reaction of quartz with high pH and Mg-rich fluids, which are the result of the dissolution of brucite from the serpentinized ultramafite. De Ruiter and Austrheim [12] focused on the similarity in structure and composition of the naturally formed Mg-silicate cement and human-made M-S-H (magnesium silicate hydrate) cement [20,21,22]. De Ruiter et al. [23] elaborated on the reaction between the high-pH fluid and quartz and highlighted the importance of deformation to enhance dissolution rates of quartz and documented by TEM work that the quartz at the contact with cement developed a µm thick amorphous zone.

In this paper, we give additional information on the Feragen M-S-H cement described by de Ruiter and Austrheim [12] and de Ruiter et al. [23]. In addition, we describe a second occurrence from Leka, Norway, reported in a master thesis by Hu [24], where M-S-H grades into M-A-S-H cement in the presence of plagioclase. Here, we add a Raman study of the cement at Leka to show that this, like the Feragen cement, is amorphous.

The motivation of this study is twofold: Firstly, this study is concerned with a poorly known but important geological process that produces new rock types within the Mg system, the feedstock for CO_2_ sequestration. Secondly, the naturally M-S-H and M-A-S-H cemented rocks may contain new reaction mechanisms and textures that may be beneficial to the industrial production of a CO_2_-neutral cement. Natural formed M-S-H and M-A-S-H cements provide information on the durability and possible mineral sources for Mg, Al and Si.

## 2. Methods

Most back-scattered electron (BSE) images and all determination of mineral and cement composition were achieved with a Cameca SX100 electron microprobe at the Department of Geology, University of Oslo. Analyses were carried out with an acceleration voltage of 15 kv, a beam current of 10 nA and a counting time of 10 s for peak and background. For most of the cement analyses, a defocused beam with a diameter of 5 µm was used to reduce evaporation of volatiles. Standards used were synthetic oxides and natural minerals. Matrix corrections followed procedures described by Pouchou and Pichoir [25]. The amount of H_2_O was estimated by subtracting the sum of the analyzed oxides from 100%.

Whole-rock geochemical analyses were performed by Actlabs Laboratories Ltd., using the lithium metaborate/tetraborate fusion ICP whole-rock and the trace element ICP/MS packages. FeO and CO_2_ were also determined by Actlab. FeO was determined through titration, using a cold acid digestion of ammonium metavanadate, and hydrofluoric acid in an open system. Ferrous ammonium sulphate was added after digestion and potassium dichromate was the titrating agent. Weight fractions of dry CO_2_ sample gas were measured by infrared absorption after decomposing 0.2 g of sample material in a resistance furnace in a pure environment at 1000 °C, using an ELTRA CW-800 (www.actlabs.com).

A Horiba Jobin–Yvon (T64000) spectrograph in micro-single mode configuration was used for the Raman experiments. A Semrock Razoredge longpass filter served to block the Rayleigh light. The entrance slit width was set to 100 microns. Combined with a 900 lines pr. mm grating, a spectrograph focal length of 640 mm and a 1024 × 256 open electrode CCD with square pixels of 2.56-micron size, a spectral width of 5.1 cm^−1^ was achieved. The spectral range from 70 to 1800 cm^−1^ was measured for every sample.

A Spectra-Physics diode pumped Millennia Pro model SJ12 laser yielding 200 mW of power at 532.1 nm was used. The beam was guided through a Pellin–Broca prism followed by a pinhole at 1.5 m distance to clean up the laser light. A set of three neutral density (ND) filters together with losses at mirrors and the built-in beamsplitter, was used for damping the power to 1.5 mW measured at the sample through an Olympus 100× objective. All Raman spectra were scale calibrated against 4-Acetamidophenol (Paracetamol) before and after the sample measurements. No attempts to control polarization were made.

To compare the change in oxide ratios caused by weathering across different initial rock compositions, we utilized simple rescaling. We divide the measured oxide ratios in the fresh and weathered parts of each sample by the oxide ratio from the fresh part. This yields a ratio that is always one in the fresh rock and different from one in the weathered parts if oxide ratios change, no matter what the initial oxide ratio of the fresh part of each sample is.

## 3. Results

### 3.1. Geological Setting and Climatic Conditions

The Feragen ultramafic complex (FUC) is located in east central Norway at an altitude between 700 and 900 m a.s.l. (Figure 1a). The area is among the coldest in Europe and located in the subarctic climate zone with monthly mean temperatures ranging between −25 and + 8 °C. The Leka ophiolite complex (LOC) is located on Leka island, Trøndelag, central Norway (Figure 1b) at a latitude of 65 °N. Leka is also situated in the subarctic climate zone with monthly mean temperatures ranging between −1 and +15 °C. Like most parts of Scandinavia, the FUB and LOC were glaciated during the Weichselian. This constrains the maximum age of the weathering and duricrust formation to <10 ka. The FUC is exposed over an area of 14 km^2^. The FUC consists of partly serpentinized harzburgite and dunite [26]. The FUC was extensively mined for chromite, starting in 1824 and terminated in 1939. The mine tailings and locally the mines shafts are coated with hydrocarbonates [5,23]. The carbonation of the mine tailings starts ca. 10 cm below the surface and is particularly well developed on the down facing side of rock fragments in the tailing [23]. On the surface at both FUC and LOC, hydrocarbonates only form under overhangs. This together with below surface carbonation of the mine tailings suggests that the hydrocarbonates are unstable in direct contact with acid rainwater. The LOC contains all the principal components of an ophiolite including a mantle section dominated by peridotite and a layered crustal sequence with alternating dunite, pyroxenite and locally plagioclase-rich (anorthositic) layers [27]. The ultramafic parts are exposed over an area of ca 20 km^2^ and are partly serpentinized [28]. Northeast of the village Solsem, South-west Leka (Figure 1b), a m thick conglomerate outcrop over 100 m^2^ is dominated by local clast from the ultramafite and gabbroic rocks, with minor quartz and k-feldspar. Bøe and Prestvik [29] interpreted the conglomerate at Solsem to be a partly lithified morainic deposited from the glaciation. Hu [24] realized that the cementing material at Solsem was similar to the M-S-H cement described from Feragen, although locally containing Al_2_O_3_.

### 3.2. Field and Microtextural Observations

The weathering of peridotite leaves a few cm thick weathering rind consisting of an outer yellow rim followed inward by blue rim while the unweathered peridotite displays a dark color [30]. The weathering rind is also seen in clasts inside the cemented moraine. Raman mapping has earlier documented that brucite present in the pristine rock was absent in the weathered rims [30]. Microtextural weathering is evidenced by the disappearance and alteration of brucite coupled with a reduction in Mg in weathering zones and an enrichment of Ni and Fe oxides in aggregates after brucite [31]. Along fault and fracture zones, weathering penetrates deep and can be followed downwards for several meters. Hydrocarbonates and pyroaurite form veins in the weathering zone of the clasts and the bedrock. The brucite contains variable amounts of FeO (Appendix A) and grades into ferroan brucite.

The cemented till at Feragen is dominated by felsic clasts and is white colored (Figure 2a–c) while the cemented moraine at Leka where the amount of ultramafic material is high displays a brownish to greyish color (Figure 2d–f). At Feragen, the M-S-H varies in thickness from 0.5 m to cm thick flakes distributed at the surface (Figure 2b). M-S-H is also developed along trenches dug in connection with the mining operation where the fluid could evaporate [12].

The deformed quartz in the till is dissolved and partly replaced by M-S-H cement along sub grain boundaries (Figure 3a–c).

Locally, the quartz is totally dissolved, and a honeycomb texture is developed (Figure 3d). De Ruiter et al. [23] demonstrated through the TEM technique that the dissolution of quartz took place through an amorphous substance. Another characteristic microtexture is etch pits on the surface of the quartz grains. K-feldspar is replaced by and aggregate of cement and µm-sized illite (Figure 3e). Locally, the aggregates after K-feldspar display flow structures with aligned illite grains. Plagioclase of varying composition from albite to bytownite (An88) are found as clasts in the cemented till at Leka.

Like quartz, plagioclase is strongly dissolved and partly replaced by cement (Figure 3f and Figure 4a–d). The cement surrounding the resorbed plagioclase is enriched in Al (Figure 4b). In the same sample, Ca-carbonate surrounds and replaces quartz grains (Figure 5a,b), suggesting that Ca released from plagioclase is consumed to form calcite. The M-(A)-S-H cement surrounds and replaces both calcite and quartz (Figure 5a,b).

Negative quartz crystals, 50 µm long and veins extending more than 200 µm into quartz grains, are filled by cement (Figure 6a,b). The location of spots identified by Raman spectra of the cement, quartz and calcite are shown in Figure 6a,b while their corresponding Raman spectra are shown in Figure 6c. The Raman spectra of the Ca-carbonate gives distinct peaks that are characteristic for calcite, but contributions from quartz are visible. The two Raman spectra for the cement are very broad and featureless and no clear peaks could be identified.

### 3.3. Composition of the Cement

The cement at both Feragen and Leka is dominated by SiO_2_-MgO-H_2_O (Appendix A). Most of the analyses from Feragen contain low Al_2_O_3_ and cluster in the central triangular part of the SiO_2_-MgO-H_2_O diagram while the cement from Leka display more variation (Figure 7a). A high-H_2_O and low-MgO variety is present at Leka (Figure 7a). Analyses from Leka contain high Al_2_O_3_ (up to 20 wt.%) adjacent to dissolving plagioclase grains. The alkalinity is low at approximately 0.1 wt.%, but the concentration locally is up to 2 and 3 wt.% of Na_2_O and K_2_O, respectively. Examples of Portland cement [32,33,34,35] are also plotted on Figure 7a,b, and it is obvious that Portland cement has a moderate content of Al_2_O_3,_ in between that is measured in the cements at Feragen and Leka, and rarely contains MgO and H_2_O. The Low H_2_O content of Portland cement is probably caused by calcination during manufacture.

### 3.4. The Composition of the Weathering Rinds

To trace the origin of the elements in the cement, we compare weathering rinds of the peridotite with its non-weathered equivalent to test the microtextural indications that peridotite release MgO during weathering. Appendix A lists the compositions for the outer (yellow) and inner (blue) weathering zones compared to the dark pristine peridotite for six samples from Feragen. For five samples, it was possible to obtain material from both the yellow and the brown weathering zones; but for one sample (FW 2.14), only the yellow zone could be separated. Also listed in Appendix A are two samples of gravel developed locally at the surface of the peridotite. The calculated MgO/SiO_2_, MgO/Fe_2_O_3_, and MgO/NiO ratios decrease form the pristine peridotite to the inner blue and outer yellow weathering rinds (Figure 8). This is in accordance with the textural evolution and suggests that MgO leaves the system. The decreasing MgO/NiO ratio in the weathered zones and in the clay gravel fraction (Appendix A) suggests that NiO does not follow MgO and remains in the regolith as reported for peridotite regolith in general [36] and for this area [30,31]. The lanthanide concentration is low (for some of the elements it is below detection limits). There is no obvious systematic change in lanthanide concentrations and patterns from the unaltered zone to the weathering rinds (Appendix A).

### 3.5. Composition of Till and Tillite

Till is produced by mechanical weathering and is assumed to produce a robust average composition of the upper continental crust [37,38,39,40]. Two samples of till from a 40 cm deep trench dug across a mudboil (Figure 9a,b) at Feragen were collected and analyzed for major and trace elements and compared with the nearby tillite (Figure 9c). The two till samples are similar in composition and characterized with elevated SiO_2_ and low Al_2_O_3_ and MgO contents (Appendix A). The nearby tillite shares the composition of the till for most elements but contains 10 wt.% of MgO and a high LOI content. The chondrite [41]-normalized lanthanide pattern (Figure 10) reveals that the till samples and the cemented till display similar patterns, with a fractionated light lanthanide (La/Gd = 5.8) and almost horizontal intermediate lanthanide and heavy lanthanide segments. The compositions including the lanthanide patterns suggest that the tillite formed from the till by addition of MgO and H_2_O. The Ni concentration in the tillite (Appendix A) is low and it appears that the Ni remains in the regolith as is also evident from the textural evolution and the compositions (Appendix A).

## 4. Discussion

### 4.1. Formation of Natural M-S-H and M-A-S-H Cement

The microtextures and the geochemistry suggest that the Mg present in the cement is derived from the weathering zone of the peridotite by dissolution of brucite. The other natural occurrence of M-S-H cement [12] is also deposited on an ultramafic complex implying that in that case, Mg also comes from the ultramafic complex. Quartz in contact with M-S-H cement both at Feragen and Leka displays dissolution textures (etch pits and negative crystals filled by M-S-H cement); and in accordance with [11], we interpret that the Si is derived from the felsic material deposited on the ultramafite. We cannot exclude that some Si is derived from dissolved Mg-silicates. However, in the absence of quartz, the Mg fluid forms hydrocarbonates and not M-S-H cement. The M-A-S-H is texturally associated with plagioclase and, like quartz, plagioclase is dissolved and replaced by cement (Figure 4b). We therefore consider that the M-A-S-H cement obtained its Al from the dissolving plagioclase. The zonation of Al in cement suggests that Al is locally derived.

The rare occurrence of natural M-S-H cement may reflect the very special geochemical environment where quartz is brought in contact with olivine-bearing rocks by glacial activity. Such an environment may be restricted to glacial terrains, where olivine and quartz are mechanically brought in contact. In magmatic systems, olivine and quartz will react at high temperature to form orthopyroxene, preventing coexisting olivine in rocks formed by magmatic processes. Glaciers and ice sheets are covering up to 30% of land surface area during glacial cycles [42]. The Snowball Earth Hypothesis suggests that ice entirely covered the earth at least twice in its history and the potential for M-S-H formation should be high. Diamictite is present through the geological record on all continents and represents lithified till with a potential for M-S-H formation. Further work will show if the few reports of natural M-S-H cement [12,13] reflect our ignorance of surface processes or if we observe a very rare geological process at Leka and Feragen.

Experimentally produced M-S-H and M-A-S-H cement are produced from amorphous substances in order to speed up reaction rates. Zhang et al. [9] documented that MSH cement can be formed experimentally by reacting brucite and silica-fume. In experimentally formed M-A-S-H cement, Al is added as metakaolin or amorphous Al_2_O_3_ [14]. In the natural cases reported here, we observe that MSH can form by reacting quartz and a high-pH fluid derived by dissolution of brucite. Raman spectra presented for experimentally produced M-S-H cement with varying Mg-Si ratios display peaks suggesting brucite [43,44]. The spectra obtained from the natural occurring cement at Leka (Figure 6c) lack distinct peaks and are typical for an amorphous substance.

### 4.2. Rate of Reaction

If the cement formed in the time span of 200 years as implied by the presence of M-S-H in the entrance to the Cr mines which were operating between 1824 and 1939, de Ruiter et al. [23] calculated a dissolution rate for quartz of 1.75 × 10^−10^ mol m^−2^ s, which is 3 orders of magnitude faster than the dissolution rate predicted by experimental studies. The microtextures of plagioclase suggest that this mineral also dissolves fast. The selective dissolution and replacement by cement along crystal planes (Figure 6a) and veins (Figure 6b) suggest that experimentally determined dissolution rates may not capture the way dissolution and reprecipitation occur in natural rock and that nature provides mechanisms that can greatly enhance dissolution rates. De Ruiter and Austrheim [23] noticed that the quartz from the tillite was strongly deformed as evidenced by undulous extinction and that the cement formed along subgrains. Bacterial activity may also enhance dissolution rates. Ferroan brucite found at both Feragen (Appendix A) and Leka [28] may suggest bacterial activity [45]. Whether the identification and cultivating of such mechanisms will speed up the kinetics to satisfy an industrial process remains an open question.

### 4.3. Durability and Strength of M-A-S-H

The natural formation of M-A-S-H is a geological process and the preservation of this cement over geological time spans suggests a high durability. In both Feragen and Leka, the cement locally develops a honeycomb texture. At Feragen, this is related to strongly deformed quartz with abundant subgrains [46]. At Leka, the honeycomb is also related to high dissolution along the crystallographic direction. Honeycomb-textured material is known to have high mechanical strength and low density [46] which are properties favorable to cement. In addition, honeycomb-textured cement has low thermal diffusivity favorable for producing energy-saving buildings [47].

### 4.4. The Magnesium Budgets

Ultramafic rocks represent a major source of MgO in the form of magnesium silicates. According to Bernard et al. [48], ultramafic rocks provide an almost unlimited reservoir of MgO, with estimated available worldwide reserves of over 10 trillion tons [49]. Assuming a thickness of 5 km for the Feragen and Leka complexes, these two complexes alone contain a MgO reserve of 4 × 10^11^ tons of MgO. To convert the Mg in silicate to a form that is reactive and applicable in an industrial process is problematic. Gardner and Sui [4] write that there is yet no known process for making magnesium oxide from natural basic magnesium silicates in an energy-efficient manner, although it may well be possible to invent one given enough support for necessary research. Mg in the form of brucite (Mg_x_Fe^2+^
_(1−x)_(OH)_2_) is a reactive substance [50] that will react with silica fume to form M-S-H cement [17]. The work of Scott et al. [51] demonstrates that brucite (Mg(OH)_2_) can be produced from olivine and serpentine through a combination of hydrolysis and acid digestion. To our knowledge, this is still not an industrial process. In nature, brucite forms from olivine-rich rocks during serpentinization according to the reaction: Mg_1.8_Fe_.2_SiO_4_ + H_2_O = (Mg,Fe)_3_Si_2_O_5_(OH)_4_ + (Mg,Fe)(OH)_2_ + Fe_3_O_4_ + H_2._ The amount of brucite formed during serpentinization is dependent on a number of parameters such as the rock type being serpentinized and the degree of serpentinization. In addition, the amount of Fe in brucite will increase its amount. We notice that both at Feragen (Appendix A) and Leka [28], the brucite contains high FeO and classifies as ferroan brucite. The brucite is intergrown with serpentine and is difficult to separate mechanically. However, the brucite dissolves in surface water to produce a Mg containing a high-pH fluid. This may represent an important way to capture the brucite. Where this fluid meets the atmosphere, it may deposit hydrocarbonates (H-M-C cement) seen as white coatings under overhangs and in mine tailings. Beinlich and Austrheim [5] measured the composition of the fluid at Feragen. The pH of mine water ranges from 9.4 to 10.6 with Mg^2+^ and Si (as SiO_2_) from 13 to 88.5 mg/L and 0 to 11.5 mg/L, respectively. The alkaline water from Leka has a pH up to 10, with Mg and silica in the same range as measured at Feragen [5]. The Mg content of the fluid at Feragen and Leka is one order of magnitude lower than fluid from other similar ultramafics [52,53]. This may mean that the measured fluid had already formed hydrocarbonates. It remains to be tested if the fluid from brucite dissolution can be collected before it deposits hydrocarbonates and if this fluid has advantages as a Mg source over the seawater that contains 52 mmol/l Mg [54].

In nature, Mg-silicate can react with CO_2_ to form magnesite. Although magnesite is not suitable for producing cement by calcination analog to Portland cement, it is not known if a fluid produced from magnesite will form H-M-S cement like our observation at Feragen and Leka. Possibly, the lower pH of such a fluid will prevent the fast dissolution of quartz and plagioclase as described above.

### 4.5. Serpentinization and Weathering

Serpentinization and weathering are used synonymously by some authors, but they are two separate processes at Leka and Feragen. While weathering occurs at and close to the surface, serpentinization occurs at higher temperatures (>200 °C). As pointed out by Goldich [55], the weathering sequence of igneous rocks is reverse of the Bowen’s reaction series, meaning that minerals formed at high temperatures like olivine weather faster than rocks formed at lower temperatures. Olivine-rich rocks are consequently more strongly weathered than most other rocks. Olivine-rich rocks contain low concentrations of Al_2_O_3_ (typically between 3 and 0.5%) and cannot form abundant Al-rich phases like kaolinite. Their natural weathering product will be quartz of clay phases consisting of Mg and Si and not necessarily brucite. A preconditioning of the peridotite by serpentinization is required to form brucite and weathering is required to dissolve the brucite and form the high pH alkaline fluid.

### 4.6. The M-(A)-S-H Cemented Till—A New Type of Duricrust

The M-(A)-S-H cemented rock at Feragen and Leka developed within till and moraine at the surface of ultramafic complexes. The M-A-S-H cemented till bears many resemblances on the members of duricrust (calcrete, silicrete). As illustrated in Figure 2a,d, the cemented till appears as cemented gravel. The first use of the term calcrete was indeed referred to as lime-cemented gravel by Lamplugh [56]. Formation of M-(A)-S-H cemented as a reaction between a precursor sediment and a fluid is a common way of forming duricrust. For example, the calcrete silcrete duricrust at Tswaane was developed within the precursor Kalahari Group sediment by laterally supplied cementing agents [57]. The duricrust formation in the Kalahari case involves the reaction between the cementing fluid and the early formed deposited duricrust. The complex relationship between calcite, quartz and M-(A)-S-H cement described above and illustrated in Figure 5a,b suggests that two fluids were involved like the duricrust described by Nash et al. [57,58]. Based on the similarities with respect to appearance and formation, we interpret the M-(A)-S-H cemented till to represent a new type of duricrust coined magsilcrete. Our knowledge of Martian and planetary crust in general is partly based on testing of crustal analogs [59]. Magsilcrete represents a new analog to be tested for remote sensing and rover data considering that mafic and ultramafic rocks must be the source material of Martian duricrust [60], where polar glacial ice sheets are present and episodic glaciation took place.

### 4.7. Formation of Natural M-(A)-S-H Cement as a Geological Process

The alteration processes of peridotite (serpentinization, carbonation and weathering) have recently gained increased attention. Serpentinization affects the petrophysical properties of rock and as such influences the geophysical imaging of the lithosphere and its geodynamic evolution. Peridotite (dunite) is the ultimate host rock for in situ CO_2_ sequestration. The reaction between olivine and CO_2_ according to the reaction Mg_2_SiO_2_ + CO_2_ = 2MgCO_3_ + SiO_2_ forms 2 mol of magnesite and 1 mole quartz and produces a rock referred to as listvenite. Weathering of peridotite or its alteration products controls the Mg cycle. Mg is a highly mobile element during weathering and is preferentially leached out of the crust compared with Si, because Si-rich minerals (alkali feldspar and quartz) are more resistant to chemical and physical weathering than mafic minerals [61].

The formation of magsilcrete as an alteration process of peridotite has up to now not been considered as an Mg reservoir during Mg-cycle modelling nor during modelling of peridotite as feedstock for CO_2_ sequestration. As Mg is consumed during magsilcrete formation, less Mg is available for Mg-carbonates. Thus, the Mg runoff from the crust is also influenced by formation of magsilcrete. Mg isotopes are applied to monitor the Mg cycle. As magsilcrete forms at low temperatures in equilibrium with hydromagnesite, nesquehonite, landsfordite, artenite and dypingite, it is also likely that it fractionates Mg isotopes. It remains to be determined how this low-temperature alteration (weathering) fractionates Mg isotopes and thus changes the foundation for modelling the Mg cycle.

## 5. Conclusions

The natural process of forming M-(A)-S-H cement involve two steps—formation of brucite by serpentinization of peridotite and subsequent dissolution of brucite during weathering to form a reactive Mg-rich high-pH fluid. The alkaline fluid dissolves quartz and plagioclase and precipitate M-(A)-S-H cement. The study of natural cement reveals that plagioclase and quartz can be used as a reactant in the production of M-A-S-H cement and that these two minerals are the main source of Al and Si in the studied cement. The study of natural cement further reveals that reacting deformed minerals enhance dissolution along subgrain boundaries, resulting in the formation of honeycomb texture cement. Honeycomb textures also form by intersection of negative quartz crystals filled by M-S-H cements. Both M-S-H and M-A-S-H are formed in glaciated areas, where glaciation deposited quartz and feldspar onto the surface of ultramafic complexes. The cemented till and moraine belong to the family of duricrusts and represent a new type of duricrust that we call magsilcrete.

## Figures and Tables

**Figure 1 materials-17-00994-f001:**
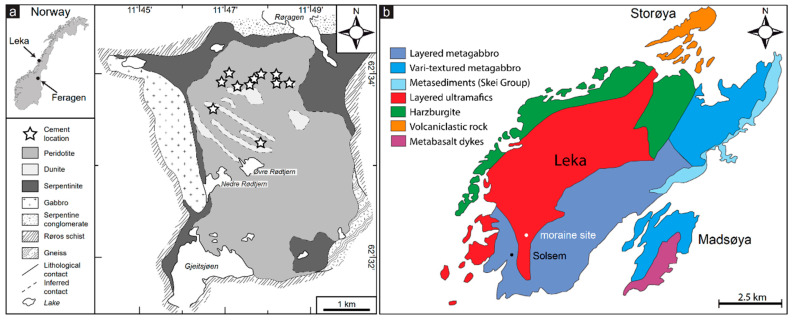
Geological maps of (**a**) the Feragen and (**b**) Leka ultramafic complexes with location of exposed M-(A)-S-H).

**Figure 2 materials-17-00994-f002:**
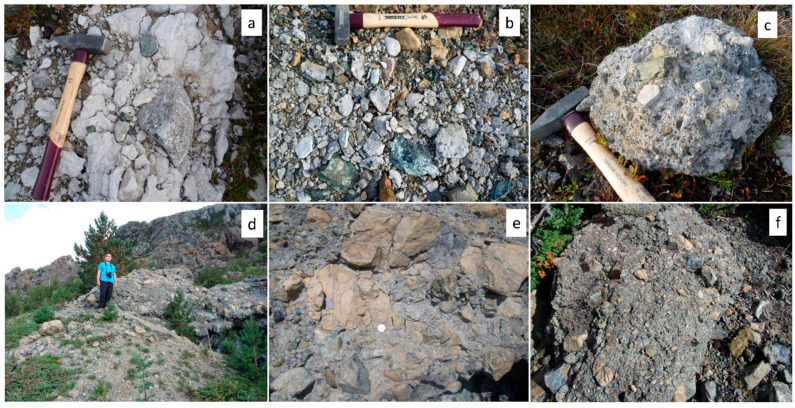
Field photos of M-(A)-S-H cemented till and moraine. (**a**–**c**) Feragen and (**d**–**f**) Leka.

**Figure 3 materials-17-00994-f003:**
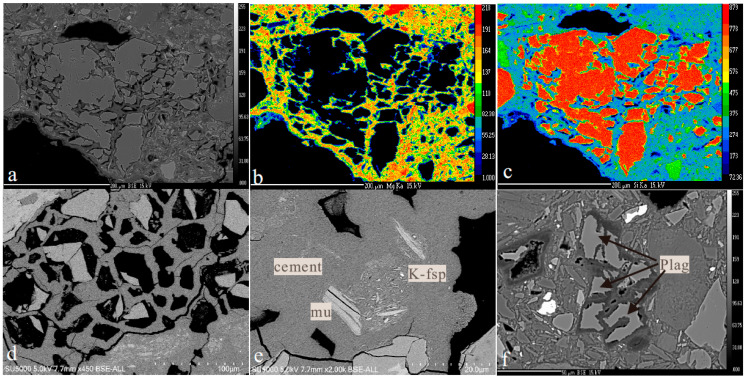
Microtextures of M-(A)-S-H cemented till. (**a**) BSE image showing large quartz grain fragmented by M-S-H cement. (**b**,**c**) Mg and Si maps of same fragmented grain. (**d**) Honeycomb microtexture after quartz. Some cells contain remnants of partly dissolved quartz. (**e**) M-S-H cement with altered K-feldspar seen as an area of illite-rich inclusions. (**f**) Partly dissolved plagioclase (An44), Leka. Veins and fingers of cement penetrate the plagioclase and result in fragmentation and replacement of the grain. Abbreviations: K-fsp: kalifeldspar, mu: muscovite, and plag: plagioclase.

**Figure 4 materials-17-00994-f004:**
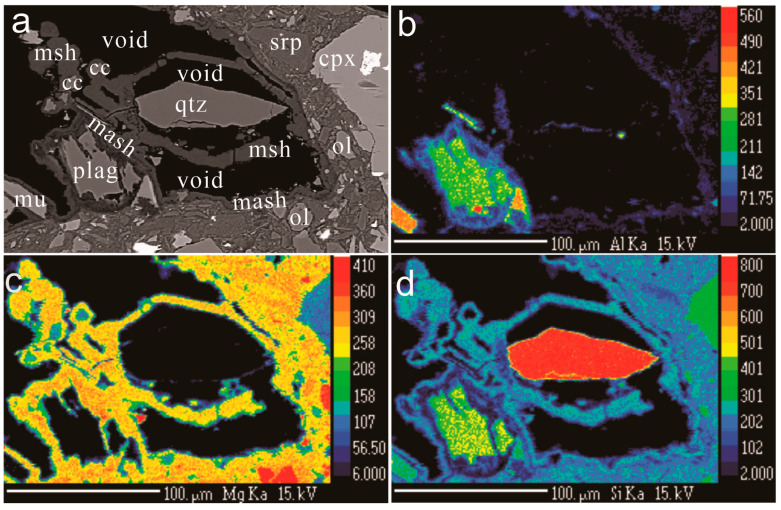
Plagioclase–quartz–cement relationships. (**a**) BSE image of adjacent grains of quartz and plagioclase surrounded by cement. (**b**) Al map over the same area. Note that the Al is present in the cement around the plagioclase and decreases away from the plagioclase grain. (**c**) Si map of the same area. The cement contains lower Si in the Al-rich zone adjacent to the plagioclase. (**d**) Mg map of the same area. Notice a thin zone of lower Mg content in the cement adjacent to the relict plagioclase grain. Abbreviations: mash: Al containing cement, plag: plagioclase, mu: muscovite, srp: serpentine, cc: calcite, cpx: clinopyroxene, msh: M-S-H cement, and void: empty space.

**Figure 5 materials-17-00994-f005:**
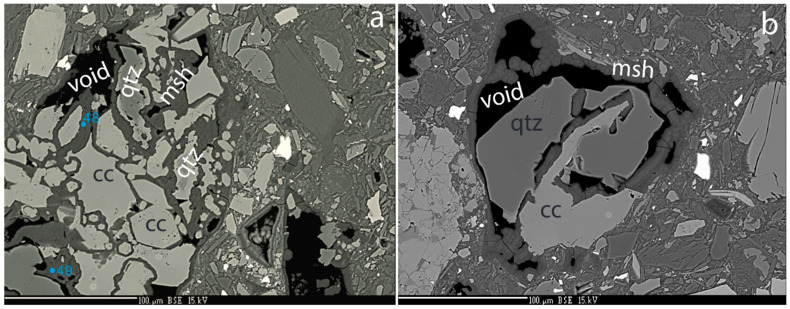
Calcite, quartz, cement relationships. (**a**) BSE image of calcite surrounding quartz grain. Veins of cement penetrated and fragment calcite grains. Abbreviations: qtz: quartz, msh: MSH cement, and cc: calcite. (**b**) BSE image of calcite vein penetrating and fragmenting quartz grain.

**Figure 6 materials-17-00994-f006:**
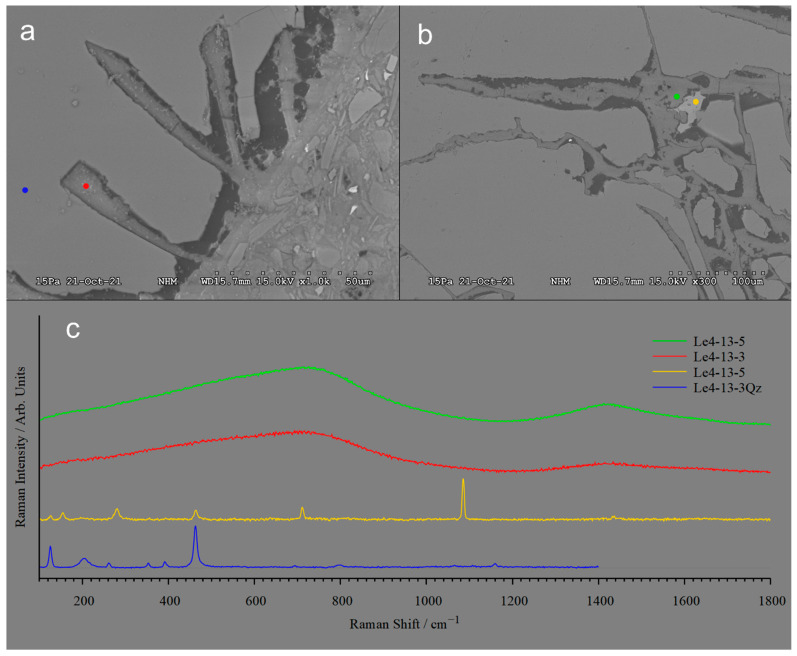
(**a**) Quartz with 50 µm long negative crystals partly filled by M-(A)-S-H cement. (**b**) Veins filled with M-(A)-S-H cement extend >100 µm into the quartz grain. In the lower right corner, a honeycomb texture has developed. (**c**) Raman spectra of cement filling, dissolved quartz, and carbonate. (**a**) The sharp peaks at 463 and 126 cm^−1^ and the broad peak at 205 cm^−1^ are typical for quartz. (**b**) This spectrum has a peak pattern localized at 154 w, 280 m, 712 m, 1085 s and 1435 w diagnostic of carbonate as situated in Calcite. In addition to these are visible residues of the quartz matrix surrounding the Calcite grain. (**c**) Both these Raman spectra are very broad and featureless, and no clear peaks could be identified, typical for an amorphous phase.

**Figure 7 materials-17-00994-f007:**
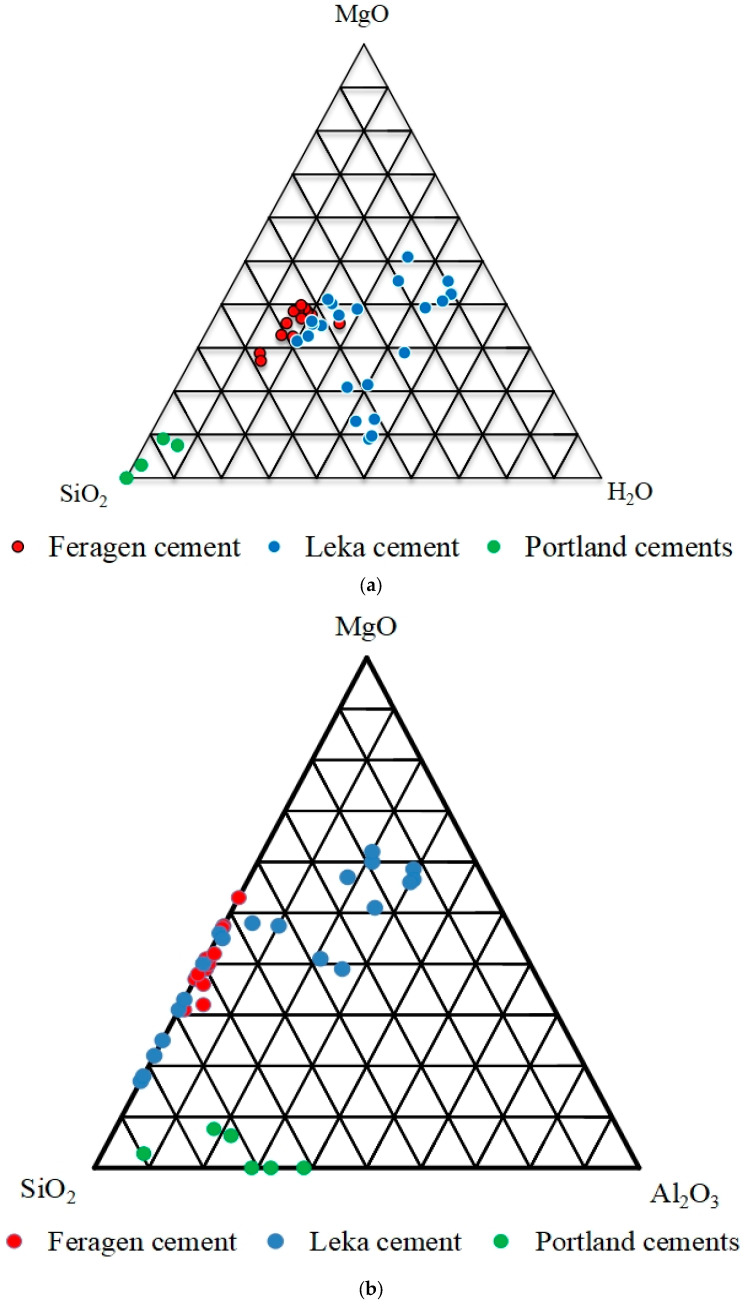
(**a**) Composition of cement from Feragen and Leka plotted in a triangular (SiO_2_-MgO-H_2_O) diagram (weight %). The samples from Feragen and some from Leka cluster in the center of diagram, while the rest from Leka are widespread, with a high H_2_O content. The composition of Portland cement is taken from [32,33,34,35]. (**b**) The same samples plotted in a triangular SiO_2_-MgO-H_2_O diagram. The samples from the Feragen plot along the SiO_2_–MgO line. The Leka samples contain variable amounts of Al_2_O_3_. The data of Portland cement are taken from [32,33,34,35].

**Figure 8 materials-17-00994-f008:**
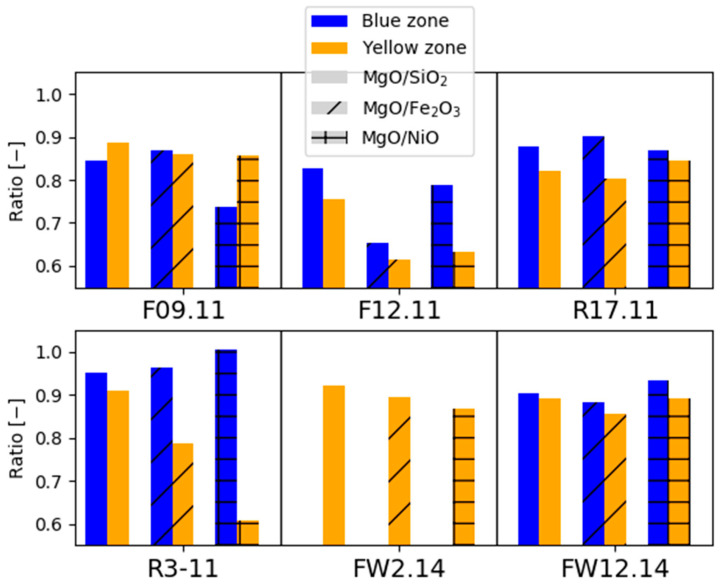
Rescaled oxide ratios across the weathering rinds for six samples from Feragen. MgO/NiO, MgO/SiO2 and MgO/FeO ratios all decrease in the traverse from the pristine peridotite where the ratio is by definition one to the intermediate (blue) and outer (yellow) zones. The data are in accordance with a loss of MgO during weathering.

**Figure 9 materials-17-00994-f009:**
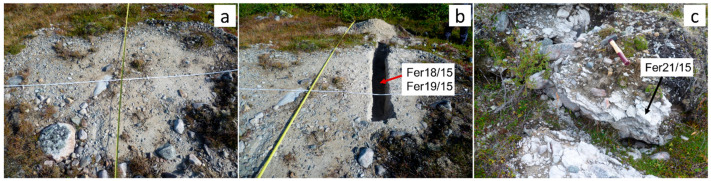
(**a**) Frost boil with till locally covers the ground at Feragen. (**b**) A trench dug through a frost boil reveals a white sand without apparent ultramafic material. (**c**) M-S-H cemented till adjacent to frost boil. A sample (FER21-15) from this outcrop was analyzed for major and trace elements and compared with till from the trench (FER18-15 and FER19-15). The red and black arrows point to the locations where the respective samples were collected.

**Figure 10 materials-17-00994-f010:**
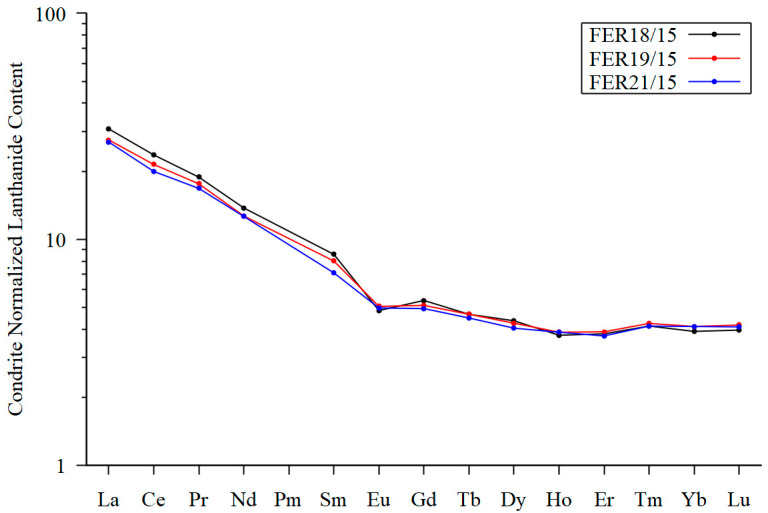
Lanthanide patterns. Chondrite-normalized lanthanide pattern for two samples of till (FER18/15 and FER19/15) compared with one sample (FER21/15) from nearby M-S-H cemented till (tillite).

## Data Availability

Data are contained within the article and Appendix A.

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
