# Peer review of "Formation of Natural Magnesium Silica Hydrate (M-S-H) and Magnesium Alumina Silica Hydrate (M-A-S-H) Cement"

_materials, 2024, doi:10.3390/ma17050994_

Round 1

Reviewer 1 Report

Comments and Suggestions for Authors

Formation of natural Magnesium Silica Hydrate(M-S-H) and Magnesium Alumina Silica Hydrate(M-A-S-H) cement

Håkon Austrheim,Depan Hu and Niels H. Andersen5

General comments:

The paper is somehow interesting, though, focusing on MSH & MASH, but naturally occurring. Although the paper provides significant data and outcomes and is well structured and written, I am not sure it will capture the interest of a large spectrum of readers of this journal. The paper might be accepted upon some minor changes as given below.

We report on“ Use the third person in your writing.

 The literature might be extended a bit more to cover most of the interconnections between the Magnesium Silica Hydrate(M-S-H) and 1 Magnesium Alumina Silica Hydrate(M-A-S-H) and their properties (mechanical and durability) of cement mortar.

Specific comments:

 “Most back-scattered electron (BSE) images and all determination of mineral and cement composition was performed with a Cameca SX100 electron microprobe at the Insti-79 tute of Geoscience, University of Oslo” I don`t think this information is important in a technical paper, and the readers are interested to know it!!! What is important for the readers is the procedure of conducting the BSE and its outcome.

Table 1 seems to be incomplete with missing information!!! The caption is also inconsequential!!

Figure 2 is showing some rocks and landscapes without a clear indication of how this can be useful for the topic, especially Figure 2d!!! This is a technical paper, not a personal Album!

 Captions of Figures 4, 5, 6, 7, 8, 9, and 10 are too long. It`s a text, not a caption! The caption should not provide detailed information.

The increase in Al2O3 take place on behave of SiO2 (Figure 7b)…..the sentence looks a bit strange!

The samples from Feragen and some of the analysis from Leka cluster in the center of diagram, while the rest of the Leka analysis show a widespread with high H2O content.” This should not be part of the caption of Figure 7. It is an explanation. It should be included in the main text.

Figure 8. the y-axis needs a title. 

Author Response

General comments:

The paper is somewhat interesting, though, focusing on MSH & MASH, but natural occurring. Although the paper provides significant data and outcomes and is well structured and written I am not sure it will capture the interest of a large spectrum of readers of this journal. Although the number of readers of a paper is difficult to foresee, you are probably right. However the scientific impact may not correlate with the number of readers. Breakthrough in science may come from interdisciplinary approaches and it only take one reader with a curiosity mind to pave out new research based on our observation. (Our group is based in geology and  I warned the editor that this paper would be geology and field related).

"We report on" use the third person in your writing. We have rewritten this. 

The literature may be extended a bit more to cover most of the interconnections between the Magnesium Silica Hydrate(M-S-H) and 1 Magnesium Alumina Silica Hydrate (M-A-S-H) and their properties (mechanical and durability) of cement mortar. We have added a reference to Bernard 2022. 

Specific comments:

"Most back-scattered (BSE) electron images and all determination of mineral and cement composition was performed with a Cameca SX100 electron microprobe at the institute of Geosciences University of Oslo". I don't think this information is important in a technical paper and the readers are interested to know it!!! What is important for the readers is the procedure of conducting the BSE and its outcome. The connection to the University of Oslo is deleted. We have chosen to keep the type of analytical instrument since this relates to the quality of the data.

Table 1 seems to be incomplete with missing information. The caption is also inconsequential!! The heading to Table 1 is rewritten.

Figure 2 is showing some rocks and landscapes without a clear indication of how his can be useful for the topic, especially Figure 2d!! This is a technical paper, not a personal Album! I hope we can keep Figure 2. We describe new geological processes  and rock types with implication for CO2 sequestration and formation of MSH and MASH cement. In geological science it is common to add data in the form of images. Such images may display microtextures but also field photos that display the dimension of the deposits etc. This is a documentation of how the MASH cemented rock looks that may help other researcher to recognize this important process in other areas.

Captions of Figures 4, 5, 6, 7, 8, 9 and 10 are too long. It's a text not a caption! he caption should not contain detailed information. We disagree, most of the texts referred to, are short descriptions of what you can see. It is in our view helpful to the reader, and without this the images are worthless. However we have deleted part of the text to figure 10 and the text to Figure 8 is new, as we have modified this figure on request from reviewer 3. 

The increase in Al2O3 take place on behave of SiO2(Figure 7b) .... the sentence looks a bit strange! This sentence has been deleted.

"The samples from Feragen and some of the analysis from Leka cluster in the center of the diagram, while the rest of the Leka analysis show a wide spread with high H2O content" This should not be part of the caption of Figure 7. It is an explanation. It should be included in the main text. We have followed your advice. and deleted this from the Figure text. This was already stated in the main text.

Figure 8. the Y-axis need a title. Figure 8 is now changed. See also comments to reviewer 3.

Reviewer 2 Report

Comments and Suggestions for Authors

The work is well written, but the purpose, motivation and context are missing in the introduction. The conclusions should refer to the initial motivations and whether these have been achieved or not. In other words, it is not clear why such research was carried out. Contextualization would make a research activity that is very far from the objectives interesting for the readers of this journal.

Author Response

The work is well written, but the purpose, motivation and context are missing in the introduction. We have added a paragraph in the introduction pointing out why we have performed this work and why the study of natural cement may bring new knowledge, not only on geological processes but may also be relevant to industrial cement production.  

The conclusions should refer to the initial motivation and weather these has been achieved or not. The conclusions have been rewritten and linked to the motivation and purpose of the study.

It is not clear why such research was carried out. Contextualization would make a research activity far from the objectives interesting for the readers of this journal.  This is a very interesting comment that reflects differences in research philosophy. A break through in science is unlikely if we all do the same thing, but may follow from curiosity research. We do not pretend that our paper is at that level of a major breakthrough, but it may contain aspect that may ignite new ideas in this field of research. One example: nature forms honeycomb textures cement. Honeycomb texture materials have properties beneficial to cement and it may be researchers out there that find this interesting and start asking how can we form this sort of texture?  

Reviewer 3 Report

Comments and Suggestions for Authors

The article presented is interesting.

First of all, avoid putting "we", write more generally. Likewise, put bibliographic references in checkmark [xx].

Reading the abstract, I had a hard time understanding the study, it needed to be revised.

In table 1, what do the 3 columns correspond to?

Furthermore, on Table 2, we see a great variability of the analyses, how to interpret?

How to ensure a stable deposit for making cement? 

Likewise for Figure 8, where the standard deviations are large? Are we lost with all these samples and we don't see the point of this approach?

The analysis could be developed on a more sustainable development scale with consideration of mineral exploitation.

In addition, it could be interesting to carry out tests on Portland cement in comparison with your samples.

It would have been interesting to put Porland cement in figures 7

The conclusion is not understandable, to be reviewed depending on the context of the study

Author Response

The article presented is interesting. Thank you for this kind comment.

First of all, avoid putting "we", write more generally. Likewise put bibliographic references in checkmarks [xx]. The sentence started with "We" is now rewritten and we have put references in checkmarks.  

Reading the abstract, I had a hard time understanding the study, it needed to be revised. We have added a paragraph on motivation of the study at the end of the introduction and hope this will make the abstract and conclusion easier to follow. 

In table 1, what do the three columns correspond to? Thank you for pointing out this mistake. We have added in the caption that they represent the composition of brucite in  samples from Feragen. 

Furthermore on Table 2, we see a great variability of analysis, how to interpret? The variability in composition of the cement must reflect local chemical domains as pointed out in the text. We see that the Al decreases away from the plagioclase in accordance with local chemical domains.

How to ensure a stable deposit for making cement? We are not sure what this mean. The chapter on the Mg budget point to sources of Mg. We have also presented estimation of MgO content in the two ultramafic bodies studied.

Likewise for Figure 8 where the standard deviations are large? Are we lost with all these samples and we don't see the point of this approach? Yes the standard deviation in Figure 8 is high. Thank you for pointing this out to us. The reason for this is that the starting material is slightly different ranging between dunite and harzburgite. We have now made a new figure 8 where we set the ratio of the oxides equal to 1 for the dark zone. This cancel out the original variation. The standard deviation is still high, but the figure demonstrate that the MgO/SiO2 and MgO/NiO is significant lower in the two weathering zones.  We also make it clear that this is work is carried out to demonstrate that the MgO in the cement is derived from the ultramafic rock. 

The analysis could be developed on a more sustainable development scale with consideration of mineral exploitation. Not clear to us what is meant here., But see our estimates of MgO content.

In addition, it could be interesting to carry out test on Portland cement in comparison with your samples. Yes, this would have been interesting, but not possible within the timeframe of this work 

It would have been interesting to put Portland cement in figures 7. Figure 7a and b now display composition of Portland cement as requested. 

The conclusion is not understandable, to be reviewed depending on the context of the study.  We have rewritten the conclusion in accordance with motivation of the study added to the introduction.

Round 2

Reviewer 3 Report

Comments and Suggestions for Authors

I accept the correction made. 

Regards

Author Response

We thank the reviewer for accepting our corrections.